# OpenReview forum: "Co-Evolving Latent Action World Models"
_ICML.cc/2026/Conference — ICML 2026 regular_

### Official Review · Reviewer_awew · 2026-03-11

**Soundness:** 3
**Presentation:** 3
**Significance:** 3
**Originality:** 3
**Overall Recommendation:** 5
**Confidence:** 3

**Summary:**

The paper introduces CoLA-World, a framework to enable joint training of latent action models and video generation models. Conventionally, these modules are trained in two stages: first, the latent action model is trained to learn actions from unannotated video data, followed by the video generation model trained to predict the future frames conditioned on the actions infered with the latent action model. The authors claim this procedure leads to redundancy, however joint training often results in representation collapse, where the pre-trained video generation model tend to ignore the noisy output of the latent action model and relies solely on its own priors. The authors propose to address this issue by a warm-up phase, where the video generation model is frozen and only emits gradients to train the latent action model. This stage is followed by full end-to-end training. Experiments demonstrate that CoLA-World achives on par or better world modelling performance compared to classical two-stage methods.

**Compliance With Llm Reviewing Policy:**

Affirmed.

**Final Justification:**

The rebuttal has addressed all my concerns and I have updated my rating accordingly.

**Key Questions For Authors:**

1) Could the authors provide a breakdown of the total TFLOPS for the joint training compared to the two-stage baseline? I would suggest to move the analysis in the appendix to the main paper.
2) Given the analysis above, what is the tradeoff between training efficiency and final downstream performance?

**Limitations:**

yes

**Strengths And Weaknesses:**

### Strengths
1) Successful implementation of the joint training of a latent action model with a pre-trained video generation model.
2) The paper provides a good intuition of why joint training typically fails, identifying representational collapse in the VQ codebook when training from scratch.
3) CoLA-World improves the performance on real-world planning compared to the two-stage baseline.
4) The paper is generally well-written and the presentaion is clear.

### Weaknesses
1) The authors claim superior efficiency of the latent action model training. However, a training step in the joint paradigm is significantly more expensive. In the two-stage baseline, the first steps only involve small networks IDM/FDM. In CoLA-World, gradients must be backpropagated through the large OpenSora world model during the whole training to update the IDM. There are some preliminary results in the appendix, but a more thorough investigation would only strengthen the paper.
2) While simulation performance appears to have improved, the linear probing loss results show only marginal gains over the two-stage baseline on several datasets (table 4). This, together with increased training costs, makes the performance improvements questionable.
3) Missing citations to early attempts on joint training of latent action and video generation models [1, 2, 3].

[1] Rybkin, Oleh, et al. "Learning what you can do before doing anything." International Conference on Learning Representations.

[2] Menapace W. et al. Playable video generation //Proceedings of the IEEE/CVF Conference on Computer Vision and Pattern Recognition. – 2021. – С. 10061-10070.

[3] Davtyan, Aram, and Paolo Favaro. "Controllable video generation through global and local motion dynamics." European Conference on Computer Vision. Cham: Springer Nature Switzerland, 2022.

---

> ### Author Rebuttal · Authors · 2026-03-31
>
> We sincerely thank the reviewer for the detailed questions regarding computational efficiency and the magnitude of our performance improvements. Building upon the preliminary analysis in Appendix D.4, we provide a deeper investigation into the total TFLOPS/compute and explicitly address the trade-off between training costs and downstream utility. We will include these analysis into the main paper in our final version.
>
> **Response to W1 & Q1: Breakdown of Computational Cost and Efficiency**\
> We appreciate the reviewer's push for a rigorous investigation into the computational overhead. We provide a detailed theoretical and empirical breakdown below.
> * **Theoretical FLOPs Breakdown**: We follow the standard $6N$ approximation for Transformer architectures ($2N$ forward, $4N$ backward per token).
> * **Per-step FLOPs Cost**: In the 2-stage baseline's LAM training phase, the encoder and decoder (both ~ 0.1B) are trained, requiring ~ 1.2B FLOPs/token. In the subsequent WM phase, the encoder (~ 0.1B) is frozen (forward only) and the OpenSora WM (~ 0.93B) is trained, requiring ~ 5.78B FLOPs/token. In CoLA-World, the warm-up and end-to-end training both require backpropagating through the WM and the encoder, taking ~ 6.18B FLOPs/token. This represents a modest ~ 6.9% increase in per-step compute.
> * **Total Theoretical Compute**: When integrating over the total training steps, comparing our Warm8K+E2E52K to the LAM30K+WM52K baseline, our joint paradigm requires roughly 10% more total theoretical FLOPs (~370.8 v.s ~336.56), depending on the exact action / video token lengths used in the isolated LAM training and WM training phases.
> * **Empirical Wall-Clock Time**: Interestingly, despite the slightly higher theoretical total FLOPs, CoLA-World is actually a little faster in wall-clock time (achieving target performance in ~ 100 hours vs. ~ 110 hours on 8 NVIDIA H200 GPUs shown in D.4). This may results from the relatively low GPU utilization in the lightweight 0.2B IDM-FDM network training, while joint training maintains consistent hardware utilization. Ultimately, the difference in FLOPs cost or wall-clock time is modest (~10%), while the joint training paradigm achieves remarkably better WM and downstream performance, as elaborated below.
>
> **Response to W2 & Q2: Magnitude of Improvements and the Overall Trade-off**\
> The reviewer questioned whether the increased per-step computational cost is justified given the "marginal" gains in linear probing. We respectfully clarify that linear probing loss alone provides an incomplete picture. When evaluated across video prediction, downstream adaptability and planning, the performance gain of CoLA-World is not marginal; it is substantial and foundational.
> * **Data Efficiency**: As shown in table 1, our WARM8K+E2E30K model already rivals the 2-stage baseline LAM30K+WM30K with a much smaller data budget. When training is extended to 60K steps (WARM8K+E2E52K), we achieve performance that outperforms the 2-stage LAM30K+WM52K baseline, clearly demonstrating superior data efficiency.
> * **Preventing Representational Collapse**: The speed of the 2-stage LAM comes at a cost of a potential shortcut representation in downstream adaptation. As detailed in Appendix D.1, when adapting to real actions, the 2-stage method suffers from severe collapse, utilizing only ~10% of its codebook. The extra per-step compute in CoLA-World is actively spent passing high-quality gradients from the WM to the LAM, guaranteeing a diverse and collapse-resistant codebook.
> * **Performance Gain in Downstream Adaptation**: Because the latent space resists collapse, real-action-based simulation improves significantly as shown in Table 3 (e.g., FVD improves from 115.45 to 93.68 on LIBERO).
> * **Doubled Utility in Planning**: Ultimately, this robust latent action space and enhanced WM prediction ability translate into a near-doubling of the relative success rate compared to the 2-stage baseline on the VP2 RoboDesk benchmark.
>
> In summary, the trade-off is highly favorable and the computation investment is fundamentally necessary for downstream success. Trading a ~10% increase in total FLOPs for a collapse-resistant representation, superior downstream prediction, and a 100% improvement in planning success is an exceptionally worthwhile investment. Furthermore, our priority in this work is to validate the feasibility and effectiveness of the co-evolutionary paradigm. With this robust foundation now established, further computational optimizations (e.g., efficient model quantization or distillation) represent highly promising directions for future work.
>
> **Response to W3**\
> We sincerely thank the reviewer for pointing out the early attempts at joint training of latent action and video generation models. We will include them in the Related Work section in our final version, highlighting how CoLA-World extends this vision to the era of large-scale, pretrained video foundation models.

---

> > ### Author Rebuttal · Reviewer_awew · 2026-04-02
> >
> > Thank you for the rebuttal. The authors have addressed all my concerns.

---

### Official Review · Reviewer_jqyT · 2026-03-12

**Soundness:** 2
**Presentation:** 3
**Significance:** 3
**Originality:** 3
**Overall Recommendation:** 4
**Confidence:** 3

**Summary:**

This paper proposes CoLA-World, a framework for jointly training a latent action model (LAM) and a pretrained video generation world model, replacing the standard two-stage pipeline where the LAM is trained first and frozen before world model training. The key insight is that the forward dynamics model in LAM and the world model do the same job (next-frame prediction), so you should just use the world model directly. The main challenge is that naive joint training collapses, where the pretrained world model ignores the random LAM signals and the codebook degenerates. The solution is a warm-up phase where the world model is frozen and only provides gradients to align the LAM, followed by full end-to-end training. The paper shows this creates a "virtuous cycle" where the world model provides better gradients to improve the LAM, and the improving LAM gives the world model a better control interface. Evaluated on robot manipulation and egocentric video data, CoLA-World matches or outperforms two-stage baselines in video prediction and downstream visual planning on VP2/RoboDesk.

**Compliance With Llm Reviewing Policy:**

Affirmed.

**Key Questions For Authors:**

- Does the same warm-up strategy work on other world model backbones without re-tuning?
- Have you run significance tests on the close results in Table 1 (e.g., OXE PSNR)?

**Limitations:**

Adequately discussed. Missing: whether the warm-up strategy generalizes to non-diffusion world model architectures.

**Strengths And Weaknesses:**

**Strengths:**
- It has a clean and very well-motivated idea, the FDM and world model are redundant, so just use the world model directly. Simple in hindsight.
- The collapse analysis is thorough and informative, the systematically testing initialization strategies to clearly explain why naive joint training fails.
- The co-evolution ablations cleanly isolate bidirectional benefit by comparing against frozen-WM and frozen-LAM variants.
- The downstream adaptation analysis is the strongest result, the two-stage baseline's codebook collapses under real-action adaptation while CoLA-World's stays healthy. This is a concrete practical advantage.

**Weaknesses:**
- Video prediction improvements are modest, e.g., 22.57 vs 22.54 PSNR on OXE is within noise. FVD gains are more meaningful but still incremental.
- Visual planning results are weak in absolute terms:13% average success is hard to call useful, even if it doubles the baseline.
- Linear probing shows nearly identical results across paradigms. The paper immediately argues probing doesn't capture full quality, which undermines including the table.
- Only one world model backbone (OpenSora) is evaluated in the paper. Given the collapse sensitivity, generalization to other architectures is unclear.
- Warm-up length (8K steps) is set empirically with no principled selection criterion- but Figure 3 shows it matters a lot.

---

> ### Author Rebuttal · Authors · 2026-03-31
>
> **Response to W1 and Q2: Video prediction improvements**\
> The close PSNR actually highlight our superior data efficiency. The 22.54 PSNR on OXE is achieved by Warm8K+E2E30K model, while the 22.57 PSNR belongs to the 2-stage LAM30K+WM30K. Achieving comparable performance with nearly half the training budget proves high data efficiency. Under the same budget (60K steps), Warm8K+E2E52K consistently surpasses the 2-stage baseline across all datasets, and outperforms the LAM30K+WM52K baseline, confirming a higher performance ceiling. Furthermore, CoLA-World outperforms the baseline across four diverse datasets in FVD, which strongly confirms statistical robustness. Notably, each result in Table 1 is the mean of 8 evaluation runs with different seeds. We will include standard deviations results in the final version.
>
> **Response to W2: Visual Planning Results**\
> We respectfully acknowledge that the absolute success rates are modest. We expand our planning evaluation to all 7 tasks in the VP2 RoboDesk benchmark across 5 seeds. We also include the reported performance from AdaWorld (a recent state-of-the-art latent-action WM). Results can be found on [our anonymous website](https://sites.google.com/view/cola-world-wm). On the 5 RoboDesk tasks evaluated by AdaWorld, their average success rate is only 12.17%, demonstrating the difficulty of the planning task. CoLA-World achieves 21.20%, nearly doubling the performance, and consistently dominating the 2-stage baseline across the board. As detailed in our response to Reviewer whxP, our method is a minimalist way to show our paradigm's effectiveness, while improvements like more expressive action adapters or advanced video backbone can be integrated to elevate the absolute planning performance in future work.
>
> **Response to W3: Linear Probing and Latent Action Quality**\
> We include the linear probing table because it is a standard evaluation benchmark. It proves that our paradigm successfully preserves basic linear correlation with real actions, achieving comparable quality to the 2-stage baseline. However, our core argument is that linear probing only partially measures the representational quality. As analyzed in Appendix D.1, the true superiority of our co-evolved LAM emerges during practical downstream adaptation: When adapting to real actions, the 2-stage LAM degenerates severely, utilizing only ~10% of the VQ codebook with a spiked max-usage metric. In contrast, our co-evolved LAM maintains a healthy codebook usage under the exact same adaptation stress. The constant supervisory feedback from the WM tutor prevents the LAM from collapse. These analysis and downstream experiments utilizing the learned LAM space prove that the co-trained LAM is more robust. We will clarify in the final version to ensure the probing results are contextualized as a basic evaluation rather than the sole metric of success.
>
>
> **Response to W4 and Q1: Alternative video generation backbones**\
> We have included the results of using another video generation model (Wan2.1-T2V-1.3B) as the WM backbone on our anonymous website in our first submission, which we mentioned in the paper in Section 5. The results show that our joint training pipeline (Warm5K+E2E40K) outperforms or achieves comparable results to 2-stage baselines (LAM30K+WM40K) even when assigned with a much lower training budget (45K v.s 70K). This demonstrates that our paradigm is backbone-model-agnostic and can benefit from advancements in foundation video generation models.
>
> **Response to W5: Warm-up length**\
> We clarify that the warm-up length choice is guided by a principled, observable criterion: the stabilization of VQ codebook health metrics. As analyzed in Figure 3, we actively monitor codebook utilization, entropy, and max-usage. The criterion to conclude the warm-up phase is the point where these metrics stabilize, indicating that the LAM has avoided initial collapse. In our experiments, these metrics stabilize around 8K–12K steps. Extending the warm-up beyond 8K is fine. We select 8K simply because it is the earliest point of convergence.
>
> **Response to the Generalization to Non-Diffusion WM Architectures**\
> We clarify that our core motivation is to adapt pretrained video generation models (which are predominantly diffusion-based models) into controllable WM. In some works of non-diffusion WM (e.g., JEPA-style LAMs, or RSSM-based PreLAR) where the WM and LAM are optimized together, they heavily rely on engineered regularizations (e.g., complex sparsity, heavy noise injection, or action-state consistency) to prevent trivial solutions. The representational collapse in our setting is caused by the capability mismatch between a powerful pretrained WM and a randomly initialized LAM. The pretrained WM simply ignores the LAM's noisy outputs. Our Warm-up strategy elegantly bridges this gap without complex constraints. We will clarify this scope boundary in the final version.

---

> > ### Author Rebuttal · Reviewer_jqyT · 2026-04-03
> >
> > I have read the rebuttal, and my main concerns have been sufficiently addressed. The authors clarified that the reported video prediction results are averaged over multiple runs, better contextualized the planning results with stronger comparative evidence, addressed the backbone generalization concern with an additional model, and explained the warm-up selection criterion more clearly. Given these, I would keep my current positive score.

---

### Official Review · Reviewer_whxP · 2026-03-13

**Soundness:** 3
**Presentation:** 3
**Significance:** 3
**Originality:** 2
**Overall Recommendation:** 3
**Confidence:** 3

**Summary:**

Instead of following the standard two-stage pipeline that first learns a latent action model and then freezes it for world model training, this work studies how to jointly train latent action models and a pretrained video-generation-based world model. To address the instability and collapse that arise in naive joint training, the paper proposes CoLA-World, which introduces a warm-up phase where the world model is frozen and only provides gradients to align the initiated inverse dynamics model before full end-to-end co-training. Experiments on large-scale action-free human egocentric and robot manipulation videos demonstrate that the proposed work matches or outperforms prior two-stage methods in latent action quality, video prediction, adaptation to real-action control, and downstream visual planning.

**Compliance With Llm Reviewing Policy:**

Affirmed.

**Final Justification:**

Some of my questions and concerns are not fully resolved, so I tend to maintain my score.

**Key Questions For Authors:**

1. Beyond short-term prediction and task success, can the proposed co-training strategy improve long-horizon consistency? Given the motivation of turning video generators into controllable simulators, it would be helpful to evaluate whether the jointly trained model yields more stable long-horizon rollouts under repeated action conditioning.
2. Why does downstream planning remain relatively low in absolute terms on difficult tasks? Although CoLA-World improves over the two-stage baseline, the practical control performance still appears limited on several challenging tasks. A more detailed discussion of the main remaining bottlenecks would strengthen the paper.

**Limitations:**

Yes

**Strengths And Weaknesses:**

Strength:
1. The paper addresses an important problem for world models and embodied AI, which is turning pretrained video generators into controllable simulators without relying on domain-specific action spaces. This is a meaningful direction, and the idea of jointly improving latent actions and world modelling could influence future work in this area. The main problem is clearly identified as the inefficiency and representational mismatch introduced by the standard two-stage pipeline. The collapse phenomenon in naive joint training is empirically demonstrated, and the proposed warm-up mechanism is simple but well-motivated by those observations.
2. Experiments are also fairly thorough, covering latent action probing, video prediction, adaptation to real-action interfaces, and planning. The results generally support the claim that co-evolution improves over the standard two-stage pipeline.
3. The paper is generally clear and easy to follow, and the ablation on co-evolution helps understand the method.

Weakness:
1. The proposed method mainly introduces a lightweight staged training strategy based on freezing and unfreezing, rather than making more substantial changes to the model architecture or optimization objective. While this simplicity is practically appealing and is enough to outperform the standard two-stage baseline, the absolute downstream planning performance remains modest on several challenging tasks. This suggests that training stabilization alone may not be sufficient, and that achieving stronger control performance may require other improvements beyond the current co-training recipe.
2. The ablations support that co-training is useful, but the claim of “mutual reinforcement” between the world model and the latent action model is still supported more by empirical results. It could be helpful by giving a deeper analysis of the insight into how the two components specifically improve one another during optimization.
3. The dataset and split description are somewhat incomplete. While the appendix clarifies that training uses the training split of the full data mixture and evaluation is conducted on validation splits, the paper does not clearly summarise the split ratio.

---

> ### Author Rebuttal · Authors · 2026-03-31
>
> **Response to W1 & Q2: The Planning Performance and the Remaining Bottlenecks**
> * **The Difficulty of the Benchmark and CoLA-World's Relative Performance Gain**\
> We agree absolute success rates are modest, as high-precision manipulation is notoriously difficult for pure visual planning. We expand our VP² RoboDesk evaluation to all 7 tasks (with more seeds) and compare with AdaWorld (a recent SOTA latent-action WM), detailing results on [our anonymous website](https://sites.google.com/view/cola-world-wm). CoLA-World consistently outperforms the 2-stage baseline and nearly doubles AdaWorld's reported performance, where AdaWorld's low performance demonstrates the difficulty in planning. Furthermore, our main contribution is to validate the feasibility and effectiveness of the co-evolutionary framework. Achieving such performance gain via a minimalist approach with no architectural or objective changes firmly proves its superiority over the 2-stage method, establishing a robust foundation for future orthogonal improvements, which we discuss below.
> * **Remaining Bottlenecks and Potential Improvements**\
> **a) The Planner**. As shown in the results on our website, increasing the number of sampled action sequences in planning, combined with a longer-trained WM checkpoint, yields noticeable performance gains. This indicates that the WM possesses the capacity for better control, but the planner struggles to efficiently search the vast action space.\
> **b) The Real-to-Latent Adapter**. Currently, we follow AdaWorld and use a 2-layer MLP to translate real actions into our learned latent action space to show the effectiveness of our pipeline in a minimal way. This simple adapter can bottleneck the control precision. Exploring more expressive models could significantly minimize adaptation errors.\
> **c) Base Video Model Capacity**. The physical grounding capability is bounded by the pretrained video backbone. Using stronger, more physically-consistent video models will elevate the baseline performance.\
> Since our method is architecture-agnostic, all the improvements can be integrated to the joint training paradigm in future work. We thank the reviewer for raising this point, and we will include detailed discussion in the final paper.
>
> **Response to W2: Deeper Analysis on Co-Evolution** \
> We thank the reviewer for encouraging a deeper analysis. As shown in Section 4.3, co-training creates a virtuous cycle of mutual promotion, where jointly optimizing the WM and LAM strictly outperforms isolated training.
> Furthermore, The downstream adaptation analysis in Appendix D.1 help us better understand the mechanism. When translating downstream real actions, the static LAM suffers severe representational collapse (codebook utilization plummets to ~10%, mapping most actions to a single code). In contrast, CoLA-World maintains a diverse, high-entropy codebook. This reflects how the two components improve one another:
> * The pretrained WM acts as a powerful tutor. Its causally sound gradient feedback forces the LAM to distribute information evenly and learn a diverse, robust action space. This supervision prevents the potential collapse.
> * The evolving LAM continuously align with the WM's manifold. Instead of forcing the WM to adapt to a static interface, this co-evolution provides a smoothly changing action landscape, unlocking the WM's predictive capacity.
>
> To conclude, this intrinsically consistent and deeply coupled system is the direct cause of our superior adaptation and planning performance. We will incorporate detailed analysis into the final version.
>
>
> **Response to W3: The Dataset Split Ratio**\
> We thank the reviewer for raising the point. For datasets that provide an official training and validation split (e.g., most OXE datasets, and the RoboDesk dataset from VP2), we adhere to the official splits. For datasets that do not provide an official split (e.g., most EgoCentric dataset, the AgiBot dataset, and the LIBERO dataset used for real-action prediction), we randomly shuffle the full dataset and hold out 5% of the trajectories as validation. We will include this clarification in our final version.
>
> **Response to Q1: Long-Horizon Video Prediction Performance**\
> We thank the reviewer for raising the point. We evaluate long-horizon autoregressive real-action video prediction across all four LIBERO suite datasets. We use the same WM checkpoints and action adapters as in Sec 4.4. Generating a long trajectory (averaging ~150 frames) requires ~25 autoregressive steps. To mitigate compounding errors, each step conditions on the 5 most recent states plus the initial anchor frame and generate the next 2 states. Results on our website show CoLA-World outperforms the 2-stage baseline across most suites and metrics. This confirms our co-trained, collapse-resistant LAM and WM successfully transfer their short-sequence prediction superiority into long-horizon rollouts. Detailed results will be added to the appendix in our final version.

---

### Official Review · Reviewer_b5SZ · 2026-03-24

**Soundness:** 3
**Presentation:** 3
**Significance:** 2
**Originality:** 3
**Overall Recommendation:** 4
**Confidence:** 3

**Summary:**

This paper proposes CoLA-world, which substitutes the forward dynamics model(FDM) in Latent Action Model(LAM) training with World Model(WM) and jointly trains the Inverse Dynamics Model(IDM) and WM. Authors identify that the failure case of jointly training lies in the inequivalent ability of a pre-trained WM and IDM from scratch, evidenced by codebook collapse. They add a warm-up phase where WM is frozen in order to align the LAM and WM. Experiments demonstrate that joint training has boost the LAM quality and WM performance on 2 in-distribution datasets, namely OXE and AgiBot, together with OOD LIBERO dataset, and co-optimizing both IDM and LAM is better than training with either fixed. Moreover, CoLA-world shows higher success rate on downstream visual planning task than two-stage WM. Extensive results show that jointly trained latent action dynamics maintain relatively high utilization and low max-usage during downstream adaptation.

**Compliance With Llm Reviewing Policy:**

Affirmed.

**Key Questions For Authors:**

1. On which dataset is Fig.4(b) experiments carried out? How is this result compared to the 2-stage training WM?

2. What would be prediction performance for warmup-only models? Since this setting resembles the 2-stage training setting and would highlight the joint training design.

**Limitations:**

Yes.

**Strengths And Weaknesses:**

Strengths:

1. The presentation of the motivation and introduction of the warm-up phase is clear and sufficient. The forward dynamics model and world model overlaps in predicting forward transition and it's natural to omit one of them and jointly trains IDM with WM.

2. The performance on downstream visual planning task achieves higher success rate. While co-optimizing eliminated the the information bottleneck posed by IDM-FDM structure, it maintains the ability to adapt to downstream real-action tasks.

3. The empirical results support the claims made and demonstrate the strength of jointly training over a two-stage one.

Weaknesses:

1. The visual planning results on ${VP}^2$ only include four tasks which is insufficient to me. AdaWorld evaluates six of them and I would suggest extensive empirical results. Intuitively receiving learning signals from pretrained WM does not necessarily strengthen IDM ability, for co-optimizing breaks the information bottleneck reserved in IDM-FDM structure.

---

> ### Author Rebuttal · Authors · 2026-03-31
>
> **Response to W1: Visual Planning Experiments**\
> We sincerely thank the reviewer for the suggestion. We extend our visual planning experiments to include all 7 tasks in the VP² RoboDesk benchmark, evaluating over more random seeds (5 seeds) for rigorous comparison. We compare our joint-training model (WARM8K+E2E30K) against the 2-stage baseline (LAM30K+WM30K), both finetuned into real-action-based WMs using the protocol from Section 4.4. We also include the reported results from AdaWorld (a recent state-of-the-art latent-action WM) for a comprehensive comparison. CoLA-World consistently outperforms the 2-stage baseline across the entire benchmark. When compared strictly on the 5 RoboDesk tasks evaluated by AdaWorld, CoLA-World also nearly doubles AdaWorld's reported performance. These extensive results firmly validate that the superior video simulation quality of CoLA-World translates into reliable visual planning performance. More results can be found on [our anonymous website](https://sites.google.com/view/cola-world-wm).
> |  | UPRIGHT BLOCK | PUSH SLIDE | FLAT BLOCK | PUSH DRAWER | PUSH RED | PUSH GREEN | PUSH BLUE| AVERAGE ON ALL 7 TASKS| AVERAGE ON 5 ROBODESK TASKS In ADAWORLD |
> | :---: |:---: | :---: | :---: | :---: | :---: |:---: |:---: |:---: |:---: |
> | JOINT | 35.33% | 4.00% | 4.00%  | 4.00%  | 19.33%  | 18.00%  | 29.33% |  16.28% | 21.20% |
> | 2-STAGE | 22.00% | 2.67% | 1.33% | 3.33% | 3.33% | 4.67% | 6.00% | 6.19% | 7.73% |
> | AdaWorld| 5.00% | 5.83% | / | / | 10.00% | 10.83% | 29.17% | / | 12.17% |
>
> (Note: AdaWorld did not report performance on Flat Block and Push Drawer)
>
> **Response to Q1: Fig.4(b) Experiments**\
> We thank the reviewer for the question. The experiments in Figure 4(b) are evaluated on the validation set of our full pretraining data mixture, which comprises the OXE, EgoCentric, and AgiBot datasets. To clarify, Figure 4(b) is not a direct comparison with the 2-stage baseline, but rather a controlled ablation study designed to verify the co-evolution circle of LAM and WM. By comparing our full Warmup + E2E paradigm against a variant where the LAM is frozen after warmup, we show that a concurrently evolving LAM provides a progressively more precise control interface, which is essential to unlocking the world model’s full predictive potential.\
> Regarding the comparison with the standard 2-stage method (LAM30K + WM): its video prediction performance consistently surpasses the frozen Warmup + WM-Only ablation shown in Fig. 4(b). In the early and mid training process, the 2-stage baseline exhibits a learning trajectory close to our standard Warmup + E2E run. In the later stages of training, it plateaus earlier and is ultimately outperformed by our joint training paradigm. This training dynamic is consistent with the final quantitative results reported in Table 1.
>
> **Response to Q2: Warmup-Only Prediction Performance**\
> We thank the reviewer for the observation. Since the warmup-only method only updates the IDM and the quantizer using the gradients backpropagated from the frozen WM, the video prediction performance significantly lags behind the joint training or the 2-stage method as the WM is not really optimized. We show the results of Warm-Only-38k below, using the same experimental protocol as in Section 4.2, and relist the performance of the corresponding joint training method. The results highlights the importance of the end2end training stage.
>
> | Dataset | Method | PSNR ↑ | SSIM ↑ | LPIPS ↓ | FVD ↓ |
> | :--- | :--- |:---: | :---: | :---: | :---: |
> | **OXE** | Joint (Warm8K + E2E30K) | 22.34 | 81.16 | 13.17 | 291.30 |
> | | Warm-Only (Warm38K)| 18.52 | 71.76 | 18.05 | 432.25 |
> | **EgoCentric** | Joint (Warm8K + E2E30K) | 23.36 | 83.41 | 13.26 | 263.57 |
> | | Warm-Only (Warm38K)| 19.67 | 71.56  | 17.67 | 432.31 |
> | **AgiBot** | Joint (Warm8K + E2E30K) | 23.64 | 85.27 | 10.22 | 189.03 |
> | | Warm-Only (Warm38K)| 20.60  | 80.38 | 13.02 | 317.94 |
> | **LIBERO** | Joint (Warm8K + E2E30K) | 23.25 | 87.05 | 10.08 | 164.86 |
> | | Warm-Only (Warm38K)| 20.08 | 81.82 | 14.12 | 395.77 |

---

> > ### Author Rebuttal · Reviewer_b5SZ · 2026-04-04
> >
> > My concern has been fully reserved after the rebuttal. The extensive results on all $7$ RoboDesk tasks further supports the benefit of joint training, while ablate on "Warmup-only" models exposes the value of the co-evolution training stage. Given these, I would maintain my positive score.

---

### Decision · Program_Chairs · 2026-04-30

**Decision:**

Accept (regular)

**Comment:**

Overall, the reviewers agree that this paper studies an important problem for embodied world models: how to effectively enable joint training of latent action models and world models. The idea is well motivated, and the paper provides useful analysis of failure cases in naive joint training, especially representation collapse. The proposed method is simple yet effective, and shows consistent improvements supported by solid experiments and ablations.

The remaining concerns are that the method is relatively incremental, with limited architectural novelty beyond a stabilized training recipe, and that improvements in absolute performance remain modest.

Nevertheless, with 3 positive and 1 negative review (and the negative reviewer not actively engaging in the rebuttal), the AC leans toward acceptance given the consistent empirical gains and practical value of this work.